# Technical Note: Intermittent reduction of the stratospheric ozone over Northern Europe caused by a storm in Atlantic Ocean

Mikhail Sofiev[1], Rostislav Kouznetsov[1,2], Risto Hänninen[1], Viktoria F. Sofieva[1]

[1]Finnish Meteorological Institute, Helsinki, 00560, Finland

[2]AM Obukhov Institute for Atmospheric Physics, Moscow, Russia

*Correspondence to*: Mikhail Sofiev (mikhail.sofiev@fmi.fi)

**Abstract.** A three-day episode of anomalously low ozone concentrations in the stratosphere over Northern Europe occurred on 3-5 November 2018. A reduction of the total ozone column down to ~200-210 Dobson Units was predicted by the global forecasts of System for Integrated modeLling of

Atmospheric coMposition (SILAM) driven by the weather forecast of Integrated Forecasting System (IFS) of European Centre for Medium-Range Weather Forecasting (ECMWF). The reduction down to 210-215 DU was subsequently observed by the satellite instruments, such as Ozone Monitoring Instrument (OMI) and Ozone Mapping Profile Suite (OMPS). The episode was caused by an intrusion of the tropospheric air, which was initially uplifted by a storm in Northern Atlantic, south-east of Greenland.

Subsequent transport towards the east and further uplift over Scandinavian ridge of this humid and low-ozone air brought it to ~25 km altitude causing ~30% reduction of the ozone layer thickness over Northern Europe. The low-ozone air was further transported eastwards and diluted over Siberia, so that the ozone concentrations restored a few days later. Comparison of the model predictions with OMI, OMPS, and MLS (Microwave Limb Sounder) satellites demonstrated the high accuracy of the 5-days forecast of the

IFS-SILAM system: the ozone anomaly was predicted within ~10 DU accuracy and positioned within a couple of hundreds of km. This episode showed the importance of the stratospheric composition dynamics and the possibility of its short-term forecasting, including such rare events.

## 1. Introduction

Quick variations (hours-to-days) of the ozone abundance in the lower stratosphere and the upper

troposphere are primarily associated with the stratosphere-troposphere exchange. Its main mechanism in extratropical regions is associated with synoptic-scale processes, in particular, extratropical cyclones (Jaeglé et al., 2017; Stohl, 2003). Attention is usually paid to intrusions of the stratospheric air into the troposphere along the descending dry-intrusion air streams of the cyclonic structure (Ebel et al., 1991; Jaeglé et al., 2017; Reutter et al., 2015; Stohl, 2001, 2003). These intrusions are estimated to be

responsible for 450-500 Tg of annual ozone import in the troposphere, which is about 10% of the ozone chemical production in the troposphere (Edwards and Evans, 2017; Olsen et al., 2013; Roelofs and Lelieveld, 2000). The uplift of the tropospheric air occurs along the ascending warm conveyor belt (WCB) of the cyclonic structure (Stohl, 2001). The dry-intrusions – WCB mechanism is responsible for 40-60%

of the intrusions in the middle latitudes over Atlantic Ocean (Reutter et al., 2015). It has been suggested

that these intrusions are quite shallow, i.e. most of the plumes do not penetrate significantly beyond the UTLS (Upper-Troposphere-Lower-Stratosphere) interface. For the stratosphere-to-troposphere (STT) intrusions, in particular, the fraction of streams reaching middle troposphere is suggested to be just 15% (Jaeglé et al., 2017).

In the above works, as well as in the earlier studies (see references in the reviews of Stohl, 2003 and

Jaeglé et al., 2017), a dominant proposition is that the intrusions related to the troposphere-to-stratosphere transport (TST) do not reach high altitudes predominantly staying within the UTLS layer where their impact on the ozone concentrations is comparatively small. Exceptions are the moist deep-convective updrafts in the tropics reaching up to 50 hPa (20 km altitude) and pollution injection up to 80-100 hPa (17-19 km) by Asian monsoon (Orbe et al., 2015). The deep penetration of the tropospheric air into the

stratosphere leads to the corresponding reduction of the ozone column. However, outside the tropical regions and the areas affected by the Asian monsoon the TST events are practically not considered.

The TST intrusions are generally less studied in the literature compare to the STT ones, which have a profound impact on the surface ozone concentrations and the tropospheric ozone budget. However, Stohl (2003) pointed out that the effect of deep intrusions may be significant and Reutter et al. (2015) estimated

that just 34% more mass is exchanged near North Atlantic cyclones for STT than for TST, average over all seasons 1979-2011.

Several other mechanisms can induce significant TST fluxes in extra-tropical regions. Powerful intrusions regularly occur along the folded tropopause at mid-latitudes. One of early modelling efforts studying this topic dates back to 1990s when the tropospheric chemistry transport model EURAD was applied to such

event and reproduced its main features under a simple assumption of a linear relationship between ozone concentration and potential vorticity (Ebel et al., 1991). A more recent diagnostic study of (Pan et al., 2009) pointed out that the association of the ozone and the thermal structures demonstrates the physical significance of the subtropical tropopause break and the secondary tropopause. However, the core of such intrusions is generally under 15 km.

The current short note analyses an unusual event that took place at the beginning of November 2018 and initially looked like a typical extratropical cyclone with sea-level pressure in the centre being just under 960 hPa. However, the WCB plume was eventually uplifted to 20-25 km and significantly affected the stratospheric ozone layer over northern Fennoscandia (60N-70N) two days later causing its intermittent reduction by as much as 30%. The episode was predicted by the SILAM model (System for Integrated

modeLling of Atmospheric coMposition) 5 days in advance and subsequently observed by the ozone-monitoring satellites.

In the following section, we present the SILAM model and outline the satellite information, which was used to confirm the event and to validate the forecasts retrospectively. The Results section presents the episode's development and evaluation of the model predictions against the satellite data. Finally,

Discussion includes a short overview of similar historical events and evaluates the significance of the current episode from the large-scale standpoint.

## 2. Forecasting model and observational data

### 2.1. SILAM v.5.6 model and input data

System for Integrated modeLling of Atmospheric coMposition (SILAM, http://silam.fmi.fi, accessed 24.01.2020 (Sofiev et al., 2015)) is an offline chemistry-transport model covering the troposphere and the stratosphere. Daily operational forecasts with SILAM v.5.6 provide the global and the regional predictions up to 5 days ahead for concentrations and deposition of 113 species. The model chemistry transformation scheme consists of: (i) the modified CBM4 mechanism (Gery et al., 1989) with updated chemistry rates, (ii) the heterogeneous inorganic chemistry of (Sofiev, 2000) expanded with marine boundary layer nitrate formation, (iii) the Volatility-Basis Set for the secondary organic aerosols, (iv) the Polar Stratospheric Cloud (PSC) formation generally following (Carslaw et al., 1995) for supercooled ternary solutions of $HNO_3+H_2SO_4$ and the formulations of the FinROSE model (Damski et al., 2007) for nitric acid trihydrate (NAT) and ice aerosols, (v) the gas-phase chemistry transformations in the stratosphere of FinROSE with an extended set of halogenated species and an updated and extended set of photolytic reactions.

Input meteorological data for the SILAM forecast are taken from the Integrated Forecasting System (IFS) of European Centre for Medium-Range Weather Forecast (ECMWF, http://www.ecmwf.int, accessed 10.12.2019). The data are used in lon-lat projection with horizontal resolution of 0.2°×0.2°×3 hr and 135 vertical levels reaching up to ~4 Pa.

Emission data are compiled from several sources. The main anthropogenic emission dataset is MACCITY (Granier et al., 2011) with shipping excluded. It is complemented with the shipping emission inventory produced with the STEAM model (Jalkanen et al., 2009, 2016; Sofiev et al., 2018). Biomass burning emission and its injection profile are calculated in real-time by IS4FIRES (http://is4fires.fmi.fi, accessed 10.12.2019, (Sofiev et al., 2009, 2013)) for aerosols and taken from the GFAS dataset (Kaiser et al., 2009) for gases. Biogenic emission is taken from the MEGAN computations (Sindelarova et al., 2014). Supplementary datasets include RETRO-aircraft (Grewe, pers.comm.), GEIA NOx from lightning (Price et al., 1997) and GEIA reactive chlorine compounds (Lobert et al., 1999) and CFCs (Cunnold et al., 1994) emissions. The emission of sea salt, wind-blown dust and DMS are computed online by SILAM (Sofiev et al., 2011). Finally, the compensating emission of $N_2O$ was estimated from the global mass budget conservation requirement and is introduced as a homogeneous constant flux from the land areas, except for Antarctica.

The SILAM forecast is run daily, 5 days ahead, with the global horizontal resolution of 0.2°×0.2° and 29 vertical levels reaching up to 5.25 Pa (mid-point of the last layer). The model does not use data

assimilation and the initial conditions are taken from the previous-day forecast. Hourly averaged 3D fields of concentrations and 2D fields of dry and wet deposition, as well as aerosol column optical thickness constitute the model output presented at the model Web site http://silam.fmi.fi (accessed 10.12.2019) in both graphical and numerical forms.

### 2.2. Satellite observations

The current study used three sets of satellite data. The total-column observations were taken from the Ozone Monitoring Instrument OMI (https://aura.gsfc.nasa.gov/omi.html, accessed 10.04.2019. (Levelt et al., 2006, 2018)) and the Ozone Mapping Profiler Suite (OMPS, https://jointmission.gsfc.nasa.gov/omps.html, accessed 10.04.2019, (Flynn et al., 2006)). Both satellites observe total ozone column over cloud-free areas and stratospheric ozone column above the clouds. Below, we present the Level 2 OMI total ozone column data with removed row-anomaly (the OMPS observations show very similar patterns). The vertical ozone profile evaluation was based on the retrievals of Microwave Limb Sounder v 4.2 (MLS, https://mls.jpl.nasa.gov/, accessed 10.04.2019, (Waters et al., 2006)). We used the MLS data from the HARMonized dataset of OZone profiles (HARMOZ, (Sofieva et al., 2013)) developed within the Climate Change Initiative of European Space Agency.

For the evaluation, the following processing has been applied to the satellite data and the SILAM results. A full space- and time- collocation was applied at hourly level, i.e. we used only those grid cells of the SILAM forecasts, for which the satellite data were available during the specific hour. The OMI / OMPS spatial resolution is higher than that of SILAM, therefore the informative satellite pixels that fell into the same SILAM grid cell were averaged. Since the columns were taken over Northern Atlantic and Scandinavia where the contribution of the lower-troposphere ozone to the total column is low, no averaging kernel was applied to the SILAM vertical ozone profile. For comparison with MLS-HARMOZ, the vertical profiles of SILAM were picked at the corresponding locations and reprojected to the HARMOZ vertical using log-interpolation in pressure coordinate.

### 3. Results

#### 3.1. Predicted evolution of the low-ozone area

According to the SILAM forecasts, the episode was started at the beginning of November 2018 in Atlantic Ocean south-east of Greenland by a strong storm (Figure 1a and Supplementary figures S1 – S7), which created a powerful updraft reaching up to nearly 15 km of altitude. Already then, this intrusion started affecting the stratospheric ozone concentrations over south-west of Norway but the reduction was just 10-15 DU (Figure 2a). The air masses were subsequently transported to the north-east and further lifted over the Scandinavian ridge gradually mixing with the ozone layer at 20-25 km altitude (Figure 1b, Figure

2ab). As a result, the area with anomalously thin ozone column (~200-210 DU) was formed over central and northern Finland (Figure 2b). In the following days, the eastward transport continued and the low-ozone air masses were transported towards Russia gradually dissolving over Siberia (Figure 2cd). The episode practically ended on 7.11.2018 but the ozone layer thickness remained somewhat low over Eurasia (230-240 DU) for a few days after (Figure 2d and the supplementary information).

In the peak of the episode, on 4 November 2018, the ozone column over Finland was 30-35% thinner than the level of 300-350 DU outside the depletion area (Figure 2).

## 3.2. Evaluation of the SILAM predictions

Evaluation of the above model predictions was performed against OMI and OMPS satellite retrievals of the ozone total column, as well as against MLS-HARMOZ vertical ozone profiles. Due to very similar patterns shown by both nadir satellites, below we discuss the OMI-based comparison. The focus was on the model ability to reproduce the absolute level of the ozone column load, as well as on accurate location of the depletion area in space and time.

The model predictions, namely the shape and evolution of the low-ozone area over Scandinavia, were confirmed (Figure 3 for 4.11.2018 and the supplementary figures S8 - S13 for the whole period). The only issue revealed by the comparison was a quite homogeneous under-estimation of the total ozone column by SILAM – within 10-20 DU over the bulk of the domain (Figure 3). This bias was also stable in time and practically did not vary throughout the episode (see the supplementary material), i.e. the anomaly of the ozone column was predicted with <10 DU error, its location was accurate within ~100 km and timeliness was captured with <1 day accuracy. Accounting for this bias, the actual ozone load was about 210-215 DU in the peak of the episode (whereas SILAM suggested it down to 200 DU), as compares to ~310-320 DU of a zonal-mean level between 60N and 80N excluding the depletion area (the corresponding SILAM mean was about 300 DU).

Considering the S1 – S7 and the corresponding S8-S13 figures, one can notice that the under-estimation of the ozone column load was somewhat stronger in the tropics than in the northern regions. This has been traced to the very low lightning emission of $NO_2$ in the input files and too intense scavenging of tropospheric ozone precursors. These resulted in low tropospheric ozone concentrations in the tropical regions thus adding ~5 DU of the under-estimation of the total column. However, these effects do not concern the current case and have been rectified in the new SILAM v.5.7 that will be put in operations in 2020.

The vertical distribution of the ozone loss on 4.11.2018 was predicted to span up to 25 km and beyond (Figure 1b). A similar effect is also seen in the MLS retrievals (Figure 4), which show that the highest ozone concentrations during the episode were predicted and observed at 22-23 km instead of usual 17-18 km. The absolute concentrations at that altitude however changed just a bit going slightly below 7 µmole

m$^{-3}$ (panel b) instead of 7.5 µmole m$^{-3}$ as the mediane level over the latitude belt outside the depletion area. One can also see that the bulk of ozone reduction occurred between the 5 km and 23 km altitude levels, but even above 25 km level the concentrations were in the lower quartile of the 60N-80N belt. This is well in agreement with the SILAM forecasts (Figure 4) and confirms an unusually strong penetration of the tropospheric air into the stratosphere. The only noticeable disagreement between

SILAM and MLS was around 15-18 km altitude, where SILAM predicted about half a µmole m$^{-3}$ lower concentrations than reported by MLS, i.e. underestimated by ~25%. However, the uncertainty of this bias is two times larger than its absolute value, which might be explained by MLS approaching the lower end of the observed altitude range. The altitude of 10 km was reached by only few MLS profiles, which nevertheless showed very good agreement.

As mentioned in the methodological section 2, the SILAM global forecasts are performed without observational data assimilation, i.e. the next forecast is started from the appropriate time step of the previous one. At a price of certain worsening of the formal scores, such as the model bias at some altitudes, this approach ensures well-balanced simulations: the quality of the forecast deteriorates only slightly over the whole predicted period (see the Supplementary material). The connection to reality is ensured by the

meteorological driver IFS, which assimilates the meteorological observations at the start of each forecast.

## 4.  Discussion

Looking into history of the OMI observations, the current episode was quite extreme although not the record-setting. In its depth on 4.11.2018, it corresponded to 0.5-th percentile of the ozone distribution in November north of 60N observed by OMI over the 12-years period of operations (2005-2017). Its strength

was a result of coincidence of otherwise usual phenomena: storm in Northern Atlantic creating the initial WCB uplift, eastwards air mass transport over the Scandinavian ridge with the additional rise, and the low solar radiation in November delaying the ozone recovery. Only three episodes, also in November (the month with the lowest ozone load in the Northern sub-polar areas), during these 12 years were stronger. The deepest decline in the subpolar region in November was in 2009 (the observed column load was

below 180 DU) followed by 2008 with minimum observed column just over 180 DU, also spanning over large area (Supplementary Figure S14). An interesting month was also November of 2012 when the median level of column load was at 300 DU instead of usual 320.DU. No evident trend in the median or minimum column loads in November in northern sub-polar latitudes were found over these years.

The overall impact of the considered episode on the large-scale atmospheric processes was small due to

its intermittent limited-area character. The reduction of the ozone amount at 12:00 4.11.2018 in comparison with the "unperturbed" level was 1.3 Tg, which is almost 30% of the layer over Finland but just 0.6% of the total ozone amount in the 60N-80N belt (205 Tg, as predicted by SILAM). However, one

has to keep in mind that during the stormy autumn/winter months quite a few cyclones have a capacity to create such depletion events.

From the health prospective, the low UV level in November in northern latitudes precluded any significant impact. For the future, the projected increase of the strength of storms can potentially make the tropospheric intrusions more significant players than was the current episode.

The climate change will probably increase the strength and frequency of such events but quantitative assessment is difficult. Indeed, as shown above, such episodes are started by strong storms. Numerous
studies summarised in IPCC Assessment Report AR5 and Special report of 1.5° global warming showed that there is a general tendency of decreasing global number of tropical cyclones and the accumulated cyclonic energy e.g., (Elsner et al., 2008), (Knutson et al., 2010), (Hoegh-Guldberg et al., 2018) and references therein. The phenomenon has been also understood from theoretical point of view (Kang and Elsner, 2015). According to these findings and future-climate projections, further decrease in cyclonic
activity is likely. However, IPCC assigned low confidence to this conclusion due to several studies reporting contradicting trends. At the same time, the number and intensity of severe cyclones and storms has increased and will probably increase further (also low confidence according to IPCC) (Knutson et al., 2013). The latter expectation is supported by e.g. statistics of strong storms in Atlantic (includes the whole of Atlantic), which shows that the number of major named storms has grown from 7 per year in 1850s to
13 in 2010s (http://www.stormfax.com/huryear.htm, visited 16.08.2019). The sharp growth started around 1990 adding almost 30% within last 30 years. Since the intermittent ozone holes will be associated with strong storms, one can expect an increase of both frequency and strength of such events in the future.

## 5. Conclusions

An episode of a strong tropospheric intrusion into the UTLS and to the middle stratosphere was predicted
by the SILAM model and subsequently observed by the ozone monitoring satellites at the beginning of November 2018. According to the model predictions, the intrusion resulted in a short (~3 days) but significant (30%, , from >300 DU down to ~200 DU) regional reduction of the total ozone column. The most-significant reduction occurred over northern Scandinavia, owing to an additional enforcement of the intrusion by the lift-up over the Scandinavian ridge.

Satellite observations of the total ozone column (OMI and OMPS) and ozone profiles (MLS) confirmed both the temporal development (within < 1 day, which corresponds to frequency of the satellite overpasses) and the spatial location of the depletion event. Absolute level of the total ozone column has been homogeneously under-estimated by ~20 DU, both within and outside the depletion area, partially due to the very low $NO_2$ emission of lightning and somewhat too strong scavenging of ozone precursors
in the troposphere. Prediction of the ozone column anomaly was within ~10 DU.

The episode corresponded to 0.5-th percentile of the OMI observations over the period 2005-2017 for the latitude belt 60N-80N in November (the month with the lowest ozone concentration in the northern sub-polar stratosphere). Despite the comparatively extreme character of the episode, its impact on the large-scale atmospheric processes and UV index at the surface was small due to intermittent character of the

ozone reduction and low level of UV radiation in Northern Europe in November. However, significance of the phenomenon can grow in the future due to increasing number of strong storms in Northern Atlantic. High accuracy of the episode prediction 5 days in advance by the IFS-SILAM system shows the possibility of prediction of details of stratospheric composition and its short-term dynamics, including such rare events.

**6. Data and model availability**

The SILAM forecasts are openly available from http://silam.fmi.fi as a week-long rolling archive. Due to large size (>2 TB per day), only a subset of the forecasts is archived over the long term. That information is available on request from the authors of the paper.

SILAM is an open-code system and can be obtained from the GitHub open repository

(https://github.com/fmidev/silam-model, visited 24.01.2020) or from the authors of the paper.

**7. Author's contributions**

MS performed the analysis of the operational forecasts and wrote the paper; RK configured the operational forecasts, participated in the analysis and writing, RH developed the new chemistry transformation scheme and participated in writing; VS performed the satellite data analysis and

participated in writing.

**8. Competing interests**

The authors declare no conflict of interests

**9. Acknowledgments**

The SILAM stratospheric modules were developed within Finnish Academy ASTREX project (grant N

139126). The work has been performed within the GLORIA project of Academy of Finland (grant N 310373). Support of ESA SUNLIT and H2020 AirQast (grant N 776361) projects is kindly appreciated.

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

**Figure 1. Panel a: MSL pressure (colour shades, hPa) and wind at ~1830 ASL (8-th hybrid model level, vectors, m s$^{-1}$) at 12:00 on 2.11.2018; Panel b: vertical ozone concentration profiles (μmole m$^{-3}$) at latitude 62N at 12:00 on 4.11.2018.**

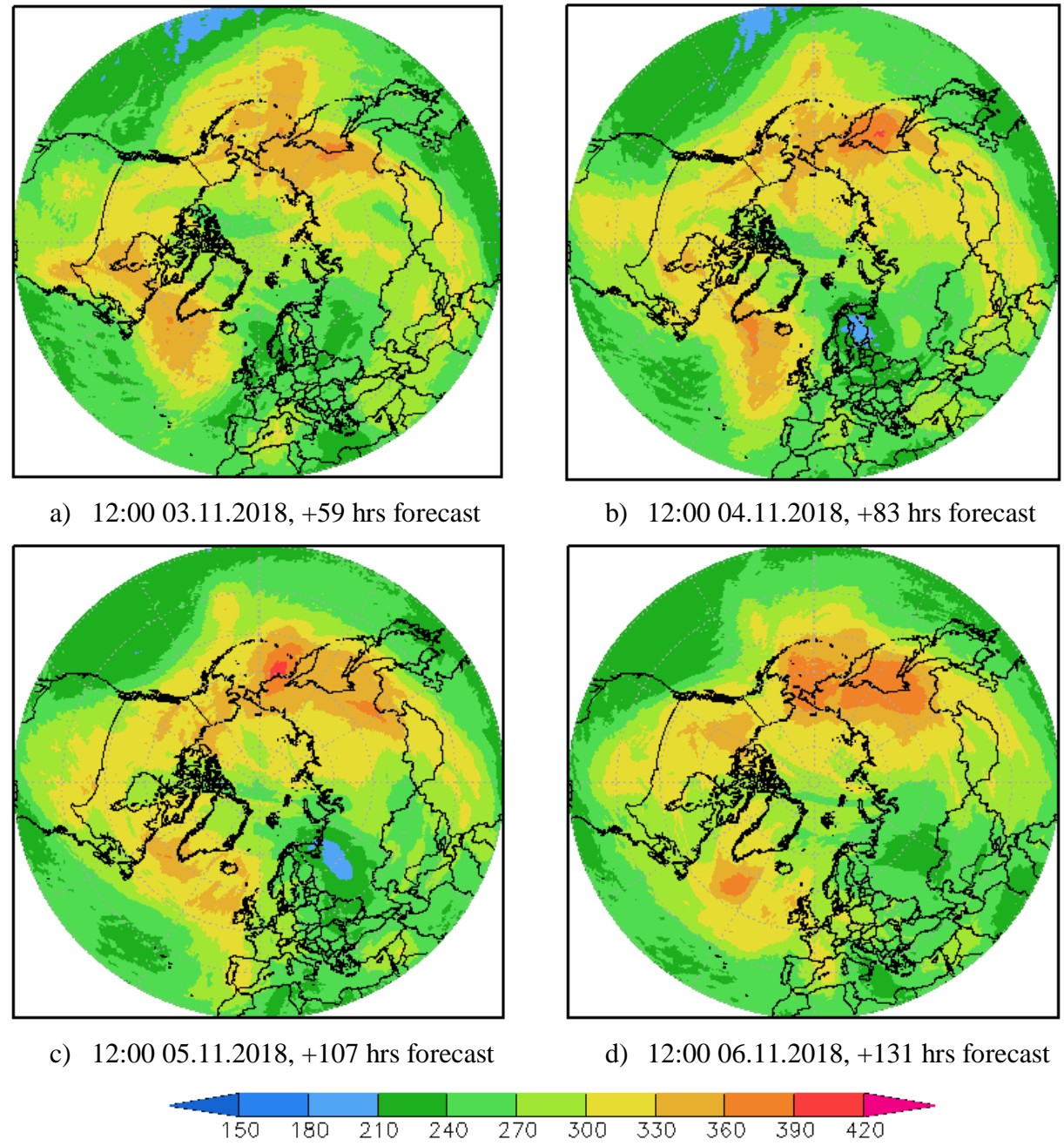

a)  12:00 03.11.2018, +59 hrs forecast        b)  12:00 04.11.2018, +83 hrs forecast

c)  12:00 05.11.2018, +107 hrs forecast       d)  12:00 06.11.2018, +131 hrs forecast

**Figure 2. Mid-day (UTC time) total ozone column in DU (Dobson units) for 3.11 – 6.11.2018 as predicted by SILAM model on 1.11.2018. Forecast length were from +59 for panel a till- +131 hours for panel d**


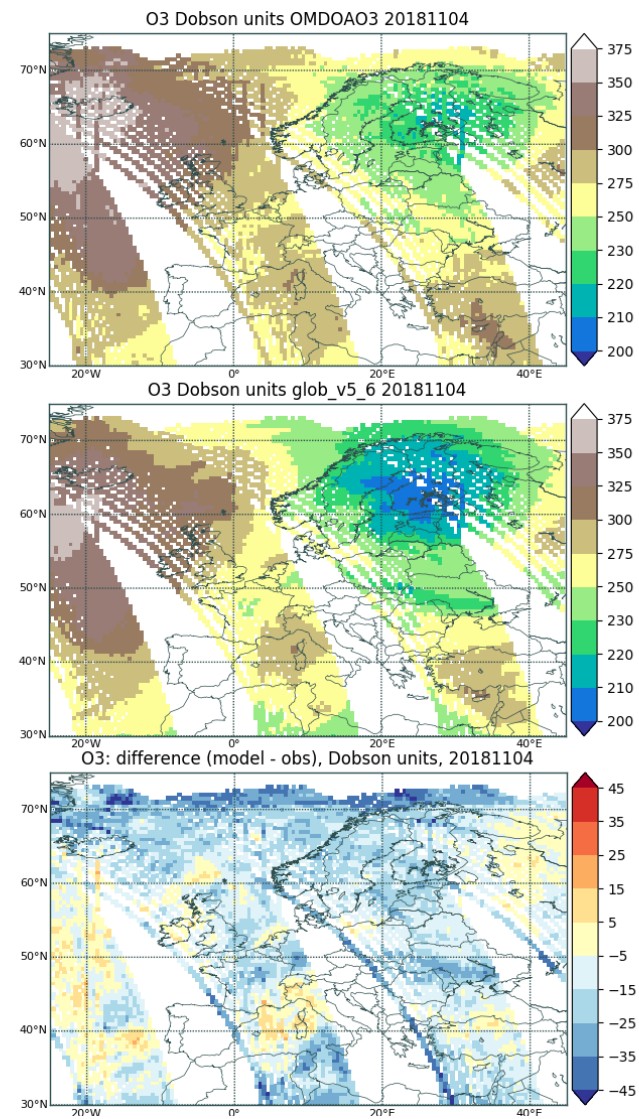

**Figure 3.** Daily-composite ozone column (DU) for 4.11.2018 observed by OMI DOAS (upper panel) and predicted by SILAM (middle panel). Only grid cells corresponding to valid OMI observations were retained in the SILAM forecast. Bottom panel: difference modelled minus observed ozone column (DU).


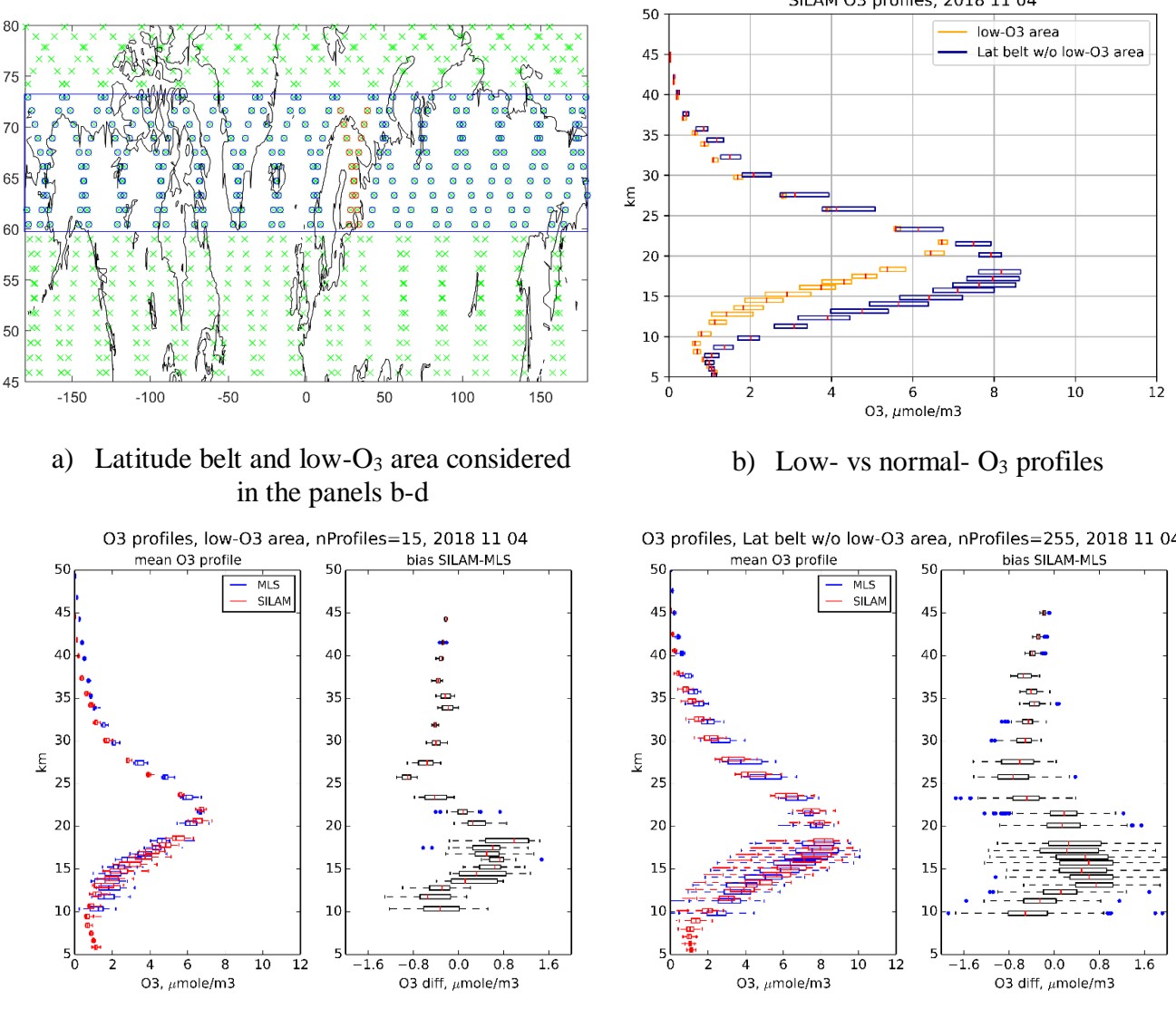

a) Latitude belt and low-O₃ area considered in the panels b-d

b) Low- vs normal- O₃ profiles

c) Low-O₃ area

d) latitude belt excluding low-O₃ area


**Figure 4. Panel a): locations of the MLS ozone profiles on 4.11.2018, the latitude belt 59N-74N and the longitudinal range 20E-40E (low-O₃ area) are highlighted. Panel b): SILAM O₃ vertical profiles predicted within and outside the low-O₃ area; panel c) MLS and SILAM ozone vertical profiles and their difference in the low-O₃ area; panel d): same as panel c but for rest of the latitude belt excluding the low-O₃ area. SILAM boxes in panels c and d are shifted upwards by 0.4 km in order to prevent overlapping pictures.**
