# Peer review of "Technical Note: Intermittent reduction of the stratospheric ozone over Northern Europe caused by a storm in Atlantic Ocean"

_Atmospheric Chemistry and Physics, 2019_

## Referee Comment (RC1) · Anonymous Referee #2 · 30 Sep 2019

In this paper, the authors describe the output of the SILAM model, which forecasts 113 species 5 days ahead. The main focus of the work is total ozone and specifically an episode of a reduction of total ozone column during early November 2018 that was forecasted by the model. The forecast was evaluated by comparing to OMI total ozone measurements and MLS ozone profiles. This technical note concludes that the SILAM model is able of high accuracy short-term forecasting of the stratospheric composition.

The paper is interesting and generally well written, even though the reader gets the sense that it was written rather hastily. There are some major issues which I think need to be addressed before final publication. These issues are in the sense major

that they are of high importance and require attention, but they are not extensive in terms of additional work required to address them. My principal concern is the limited discussion on the quantitative comparison between the model and the observations of OMI and, even more importantly, MLS. For example, a figure showing the differences (or percentage differences) between SILAM and MLS ozone profiles is missing. This will add to the value of this work, showing the capabilities of the model to capture the variations of ozone horizontally (using comparisons to OMI, which are already included in the manuscript) and vertically (using comparisons to MLS profiles). To my opinion, the paper must be adequately revised before publication.

Major Issues

1. Throughout the paper, total ozone is expressed in Dobson Units, $\mu$mole m-3 and mole m-3. It is very important that the authors conclude in one of these units and change the figures and manuscript according to it. I would suggest using Dobson Units.

2. The methodology of the comparisons should be briefly mentioned in the abstract.

3. There is a significant issue about the quantitative evaluation of the SILAM model. In Figure 3 and Figures S8-S12, the differences between model and satellite are always spanning from +10 DU to -30DU. Why is the scale of these figures so large? These figures need to be produced again with a scale e.g. +10 to $\sim$-50 DU and with a finer analysis, so that the reader will be able to easily see the areas with high differences.

4. Section 2.2: This section should be enriched with information concerning the algorithm that was used for the retrieval of ozone from the satellite measurements and on the ways this could this affect the difference found between the model estimations and the satellite measurements. Some more information on the collocation methodology are also missing. Remember that the reader must be able to reproduce your scientific methodology, or at least understand it.

5. Figures S1 – S7 show a latitude belt between ∼10-30 degrees N with total ozone below 210 DU. This is a very low estimation, since in that latitude belt total ozone values are rarely below 220 DU during November, usually ranging between 220 and 280 DU (based on OMI and TROPOMI/S5P total ozone measurements). This underestimation and its effect on the model's overall output should concern the authors. Please discuss and correct if possible. Moreover, all figures S1-S7 have the same legend.

6. Page 4, lines 6-7: Some references or an explanation (in case that the OMI data are used to extract this conclusion) to support the "usual level of 300-350 DU", are needed.

7. Page 4, line 18: Does this mean that you actually corrected the SILAM model itself or its output for the bias? This is not clear here.

8. Page 4, line 19: Where did this "∼310-320 DU" result from?

9. Most of the "Discussion" section is just a second introduction. This is not a discussion of the work done and its outcomes and it should be moved to the respective section. To my opinion, Section 4 is the "discussion" section and it should be renamed.

10. The "Conclusions" section is very short, it looks like it is written in the form of bullet points and lacks coherence. It needs to be restructured and should focus on the temporal and spatial quantitative accuracy of the prediction of such phenomena by the model. The differences and the bias between model and satellite measurements should be summarized and commented here. This is also the section to point out the novelties and significance of this work and its contribution to our knowledge about incidents like this.

Minor Issues

1. page 1, line 16: "The high accuracy . . .."

2. Page 2, line 16: The second sentence of this paragraph should be rephrased. It is not clear what this means.

3. page 3, line 17: Please rephrase as follows "The current study used three sets of satellite data: from Ozone Monitoring Instrument OMI. . ."

4. Page 4, line 14: Please rephrase, e.g. as follows "The model predictions, namely the shape. . ..".

5. Page 4, line 27: Please give the number of the section you are referring to.

6. Figure 1a shows the "Meteorological situation" (please rephrase that) for 2.11.2018 and the figure legend states that this figure refers to 3.11.2018. Please correct this.

7. Page 5, lines 5-7: The sentence "Its strength. . . ozone recovery" discusses the episode under study, while the rest of the paragraph describes the historical record of total ozone during November in the latitude belt above 60 degrees N. This sentence should be slightly rephrased (e.g. "The episode of November 2018 was a result. . .") and placed at the end of the paragraph.

8. Page 5, lines 29: The first sentence of the paragraph should be rephrased, e.g. "The bulk impact of the episode under study. . .".

9. The word "bulk" is too frequently used. Please use another synonym, instead.

---

## Referee Comment (RC2) · Anonymous Referee #3 · 1 Oct 2019

General comments

This technical note is a case study about an episode of reduced stratospheric ozone over Northern Europe caused by the intrusion of tropospheric air, which could be forecasted 5 days ahead by the SILAM model. The event has been validated with satellite observations from three different sensors. This study is a scientifically interesting case, which was conducted appropriately by the the authors. The structure and presentation of the results of this paper need to be improved:

1. Structure of the manuscript: The structure of the manuscript needs to be changed in my opinion. The discussion section is disappointing. It contains mainly descriptions

about other events and an outlook of what might or might not happen due to climate change in the future. A discussion of the model forecast results and evaluation is missing in this section. I would thus suggest a re-structuring of the manuscript: 1. Introduction, 2. Model and observations 3. Results, 4. Conclusions. Sections 3 and 4 would then go into the results section. I would shift the first part of the discussions section (about the general character of TST events) to the introduction. The second paragraph could move to the conclusion section (in a somewhat condensed way).

2. Section 4: Evaluation of the SILAM predictions: In section 3 the evolution of the event is nicely described according to the forecasts. I think the validation results should be presented in a similar, more detailed way. It is merely a few sentences that describe the total of the results. However, this is the most interesting part! It would be good to know whether the underestimation of total ozone in the model was present before and after the event as well or just during the event. Is there an explanation for this underestimation in the model? Also, more quantification of errors of some kind (e.g. table of biases) would be nice.

Specific comments:

1. Page 2, line 31: could you explicitly name the ECMWF product which was used? Sure it is not 137 levels?

2. Page 2, line 8: "However, outside the tropical regions and areas affected by the Asian monsoon the TST events are practically not considered." Is that because they do occur only randomly or because no one has investigated this before or both?

3: Page 2, line 14: In the following section, we present the SILAM model, which forecasted the episode 5 days in advance Maybe revise the sentence. Not the SILAM model is presented but the results of the SILAM model forecasts in combination with IFS meteorological input.

24. Page 3, line 15: What do you mean by "the bulk of the domain?"

4. Page 3, line 17: I think this paragraph is a bit tangled. I would suggest specifying the three satellite data sets first (OMI, OMPS and MLS) and put the rest later. 5. Page 3, line 26: 3 Predicted evolution of the low-ozone area I would shift this part in the results section. 6. Page 5, line 32: Please reformulate, it is not clear what this sentence means.

Technical corrections

1. Title: Maybe change to: Intermittent reduction of the stratospheric ozone over Northern Europe caused by a storm in the Atlantic Ocean?

2. Page 1, Line 9: Please reformulate!

3. Page 1, Line 12: Change to: The episode was caused by the intrusion of tropospheric air, which was initially uplifted by a storm in the Northern Atlantic, south-east of Greenland.

4. Page1, Line 13: Change to: . . . over the Scandinavian ridge . . .

5. Page 2, line 1: Change to: over the Atlantic Ocean

6. Page 2, line 1: I would suggest: The majority of studies on the . . . .

7. Page 2, line 3: Maybe change to: in the above mentioned studies . . . Rest of the sentence needs to be reformulated.

8. Page 2, line 3: Change to: by the Asian monsoon

9. Page 2, line 8: Better: leads to a corresponding reduction

10. Page 2, line 22: better: . . .and providing global and regional forecasts up to 5 days ahead for 113 species.

11. Page 2, line 23: CBM4: explain abbreviation

12. Page 2, line 31: Better: . . .of the European Centre for Medium-Range Weather Forecast

13. Page 3, line 2: Change to: Shipping emissions

14. Page 3, line 3: Maybe better: Biomass burning emissions and injection profiles. . .

15. Page 3, line 5: Biogenic emissions are. . .

16. Page 3, line 6: (Grewe, pers.comm.) Really no other citation available?

17. Page 3, line 14: web site

19. Page 3, line 17: The current study used. . .or this study used. . .

20. Page 3, line 12: Change to: The main focus of the evaluation was set on the model's ability to reproduce the absolute level of the ozone column load, as well as on the accurate location of the depletion area in space and time.

21. Page 3, line 15: Change to: underestimation

22. Page 3, line 15: What do you mean by "the bulk of the domain?"

23. Page 3, line 22: Please reformulate that sentence

24. Page 4, line 6: Change to: a storm in the Northern Atlantic creating the initial. . .

25. Page 4, line 6: Maybe better: air mass transport eastwards over

26. Page 4, line 9: please reformulate that sentence

27. Page 4, line 14: Better: In the model predictions. . ..

28. Page 5, line 11: Change to: . . .was found. . .

29. Page 5, line 24: Better: The impact of the Episode

30. Page 5, line 29: May be the author meant "perspective" instead of "prospective"?

31. Page 6, line 21: Better: the intermittent character of the ozone. . .

In supplement: Please change in figure captions Dubson to Dobson

---

## Author Comment (AC1) · 3 Dec 2019

Author response to the referee comments to Sofiev et al,

First of all, we would like to express our deep gratitude to the referees for their thorough work and detailed comments, which helped us improving the manuscript. We closely followed the recommendations while preparing the revised version of the paper. The reviewer's comments and our responses are presented below.

Referee 2

My principal concern is the limited discussion on the quantitative comparison between

the model and the observations of OMI and, even more importantly, MLS. For example, a figure showing the differences (or percentage differences) between SILAM and MLS ozone profiles is missing. This will add to the value of this work, showing the capabilities of the model to capture the variations of ozone horizontally (using comparisons to OMI, which are already included in the manuscript) and vertically (using comparisons to MLS profiles).

A. The figure comparing the MLS and SILAM profiles has been added and the evaluation discussion expanded, also accounting for the request of the Reviewer 3.

Throughout the paper, total ozone is expressed in Dobson Units, $\mu$mole m-3 and mole m-3. It is very important that the authors conclude in one of these units and change the figures and manuscript according to it. I would suggest using Dobson Units.

A. The units have been harmonised: column load is now in Dobson Units whereas concentration is in $\mu$moles m-3.

2. The methodology of the comparisons should be briefly mentioned in the abstract.

A. Added

3. There is a significant issue about the quantitative evaluation of the SILAM model. In Figure 3 and Figures S8-S12, the differences between model and satellite are always spanning from +10 DU to -30DU. Why is the scale of these figures so large? These figures need to be produced again with a scale e.g. +10 to âĹij-50 DU and with a finer analysis, so that the reader will be able to easily see the areas with high differences.

A. The figures have been redrawn

4. Section 2.2: This section should be enriched with information concerning the algorithm that was used for the retrieval of ozone from the satellite measurements and on the ways this could this affect the difference found between the model estimations and the satellite measurements. Some more information on the collocation methodology are also missing. Remember that the reader must be able to reproduce your scientific

methodology, or at least understand it.

A. We expanded the outline of processing of the ozone products of OMI used in the comparison. More details are also provided on the colocation method.

Figures S1 – S7 show a latitude belt between âĹij10-30 degrees N with total ozone below 210 DU. This is a very low estimation, since in that latitude belt total ozone values are rarely below 220 DU during November, usually ranging between 220 and 280 DU (based on OMI and TROPOMI/S5P total ozone measurements). This underestimation and its effect on the model's overall output should concern the authors. Please discuss and correct if possible. Moreover, all figures S1-S7 have the same legend.

A. Yes, it was our concern indeed. This effect has just been traced to the problems with the lightning emission and some missing chemical reactions in the troposphere. Together with sub-optimal scavenging in tropics, they were causing a significant under-estimation of the tropospheric ozone in tropical regions. A new version of SILAM is gradually emerging with better skills in the tropical troposphere. However, the issue has little effect on the stratospheric concentrations and is fading out outside the trop-ics. Therefore, it does not affect the results of the current paper. We added a brief discussion in the Annex, next to the figures S1-S7, which captions were also corrected.

6. Page 4, lines 6-7: Some references or an explanation (in case that the OMI data are used to extract this conclusion) to support the "usual level of 300-350 DU", are needed.

A. Clarification added

7. Page 4, line 18: Does this mean that you actually corrected the SILAM model itself or its output for the bias? This is not clear here.

A. No, the model results are presented as they were. The sentence has been clarified

8. Page 4, line 19: Where did this "âĹij310-320 DU" result from?

A. This is just a "typical" level of ozone load in the 60-70ïĆř outside the region affected

by the depletion – the yellow-shaded areas in Figure 2.

9. Most of the "Discussion" section is just a second introduction. This is not a discussion of the work done and its outcomes and it should be moved to the respective section. To my opinion, Section 4 is the "discussion" section and it should be renamed.

A. The corresponding part of section 4 has been moved to Introduction, also accounting for the restructuring request of the Referee 3.

10. The "Conclusions" section is very short, it looks like it is written in the form of bullet points and lacks coherence. It needs to be restructured and should focus on the temporal and spatial quantitative accuracy of the prediction of such phenomena by the model. The differences and the bias between model and satellite measurements should be summarized and commented here. This is also the section to point out the novelties and significance of this work and its contribution to our knowledge about incidents like this.

A. The conclusion section has been reviewed, also following the restructuring request.

Minor Issues 1. page 1, line 16: "The high accuracy . . .." 2. Page 2, line 16: The second sentence of this paragraph should be rephrased. It is not clear what this means 3. page 3, line 17: Please rephrase as follows "The current study used three sets of satellite data: from Ozone Monitoring Instrument OMI. . ." 4. Page 4, line 14: Please rephrase, e.g. as follows "The model predictions, namely the shape. . ..". 5. Page 4, line 27: Please give the number of the section you are referring to. 6. Figure 1a shows the "Meteorological situation" (please rephrase that) for 2.11.2018 and the figure legend states that this figure refers to 3.11.2018. Please correct this. 7. Page 5, lines 5-7: The sentence "Its strength. . . ozone recovery" discusses the episode under study, while the rest of the paragraph describes the historical record of total ozone during November in the latitude belt above 60 degrees N. This sentence should be slightly rephrased (e.g. "The episode of November 2018 was a result. . .") and placed at the end of the paragraph. 8. Page 5, lines 29: The first sentence of the paragraph

should be rephrased, e.g. "The bulk impact of the episode under study. . .". 9. The word "bulk" is too frequently used. Please use another synonym, instead

A. Thank you for the detailed editions! We introduced the corrections

Referee 3 Structure of the manuscript: The structure of the manuscript needs to be changed in my opinion. The discussion section is disappointing. It contains mainly descriptions about other events and an outlook of what might or might not happen due to climate change in the future. A discussion of the model forecast results and evaluation is missing in this section. I would thus suggest a re-structuring of the manuscript: 1. Introduction, 2. Model and observations 3. Results, 4. Conclusions. Sections 3 and 4 would then go into the results section. I would shift the first part of the discussions section (about the general character of TST events) to the introduction. The second paragraph could move to the conclusion section (in a somewhat condensed way).

A. The paper structure has been reviewed, also accounting for the comments 9 and 10 of the Referee 2. Namely, part of discussion has been moved to introduction whereas section 3 and 4 became the new Results and discussion section.

2. Section 4: Evaluation of the SILAM predictions: In section 3 the evolution of the event is nicely described according to the forecasts. I think the validation results should be presented in a similar, more detailed way. It is merely a few sentences that describe the total of the results. However, this is the most interesting part! It would be good to know whether the underestimation of total ozone in the model was present before and after the event as well or just during the event. Is there an explanation for this underestimation in the model? Also, more quantification of errors of some kind (e.g. table of biases) would be nice.

A. The evaluation has been extended, also following the request of Referee 2. We expanded the MLS comparison of the vertical ozone profile and highlighted that the model skills (in particular, its bias) were not related to the episode but rather reflecting the somewhat too low oxidation capacity of the current SILAM chemistry scheme,

especially in the troposphere.

---

## Author Response (AR2)

Dear Editor,

Thank you very much for handling the review process and comments to the paper! We also would like to thank once again the reviewers for their efforts and comments.

The final editions entirely concentrated on the language and style of the presentation are shown below.

Thank you once again on behalf of all co-authors,

Mikhail Sofiev

[revised manuscript text omitted]